# Comparison of 46 Cytokines in Peripheral Blood Between Patients with Papillary Thyroid Cancer and Healthy Individuals with AI-Driven Analysis to Distinguish Between the Two Groups

**DOI:** 10.3390/diagnostics15060791

**Published:** 2025-03-20

**Authors:** Kyung-Jin Bae, Jun-Hyung Bae, Ae-Chin Oh, Chi-Hyun Cho

**Affiliations:** 1Department of Medicine, Korea University College of Medicine, Seoul 02841, Republic of Korea; qorudwls10@korea.ac.kr (K.-J.B.); reecebaejun@naver.com (J.-H.B.); 2Department of Laboratory Medicine, Korea Cancer Center Hospital, Seoul 01812, Republic of Korea; 3Department of Laboratory Medicine, College of Medicine, Korea University Ansan Hospital, Ansan-si 15355, Gyeonggi-do, Republic of Korea

**Keywords:** papillary thyroid carcinoma, healthy, cytokines, AI, peripheral blood

## Abstract

**Background**: Recent studies have analyzed some cytokines in patients with papillary thyroid carcinoma (PTC), but simultaneous analysis of multiple cytokines remains rare. Nonetheless, the simultaneous assessment of multiple cytokines is increasingly recognized as crucial for understanding the cytokine characteristics and developmental mechanisms in PTC. In addition, studies applying artificial intelligence (AI) to discriminate patients with PTC based on serum multiple cytokine data have been performed rarely. Here, we measured and compared 46 cytokines in patients with PTC and healthy individuals, applying AI algorithms to classify the two groups. **Methods**: Blood serum was isolated from 63 patients with PTC and 63 control individuals. Forty-six cytokines were analyzed simultaneously using Luminex assay Human XL Cytokine Panel. Several laboratory findings were identified from electronic medical records. Student’s *t*-test or the Mann–Whitney U test were performed to analyze the difference between the two groups. As AI classification algorithms to categorize patients with PTC, K-nearest neighbor function, Naïve Bayes classifier, logistic regression, support vector machine, and eXtreme Gradient Boosting (XGBoost) were employed. The SHAP analysis assessed how individual parameters influence the classification of patients with PTC. **Results**: Cytokine levels, including GM-CSF, IFN-γ, IL-1ra, IL-7, IL-10, IL-12p40, IL-15, CCL20/MIP-α, CCL5/RANTES, and TNF-α, were significantly higher in PTC than in controls. Conversely, CD40 Ligand, EGF, IL-1β, PDGF-AA, and TGF-α exhibited significantly lower concentrations in PTC compared to controls. Among the five classification algorithms evaluated, XGBoost demonstrated superior performance in terms of accuracy, precision, sensitivity (recall), specificity, F1-score, and ROC-AUC score. Notably, EGF and IL-10 were identified as critical cytokines that significantly contributed to the differentiation of patients with PTC. **Conclusions**: A total of 5 cytokines showed lower levels in the PTC group than in the control, while 10 cytokines showed higher levels. While XGBoost demonstrated the best performance in discriminating between the PTC group and the control group, EGF and IL-10 were considered to be closely associated with PTC.

## 1. Introduction

Thyroid tumor is the most common endocrine malignancy, with its incidence having increased rapidly over the past several decades [1,2]. Papillary thyroid cancer (PTC), a subtype of differentiated thyroid cancer (DTC), accounts for the majority of thyroid cancers and generally has a favorable prognosis [2]. The initiation and differentiation of these thyroid tumors are significantly influenced by the tumor microenvironment, where cytokines and chemokines are secreted to facilitate intercellular communication [3]. These molecules play extensive roles in tumor proliferation, differentiation, and survival [4,5]. For example, IL-1α has been reported to promote proliferation in PTC via the activation of Ca^2+^ channels [6], whereas IL-1β is known to exert a potent antitumor effect in PTC, inhibiting both proliferation and invasiveness [7]. Although the specific mechanisms of each cytokine or chemokine are not yet fully understood, their potential utility in cancer diagnosis has been increasingly highlighted in various studies [4,8].

In this context, recent studies have analyzed some cytokines in patients with PTC, but simultaneous analysis of multiple cytokines remains rare [4]. Moreover, most of these studies compare patients mainly with those having thyroiditis or focus solely on interleukins (IL), with only a few studies distinguishing patients with PTC and control by analyzing multiple cytokines [9,10]. Therefore, simultaneous assessment of multiple cytokines in patients with PTC and healthy individuals is crucial not only for understanding the mechanisms of PTC development and the cytokine characteristics in PTC groups but also for developing therapeutic agents [4,9]. Especially given that the diagnosis of PTC primarily relies on an invasive method such as fine-needle aspiration cytology (FNAC) [11], discovering that certain cytokines can effectively distinguish between patients with PTC and healthy individuals would enable their use in screening tests. This finding would also contribute to the development of treatments other than surgical removal, which is the current treatment of choice [12].

Meanwhile, artificial intelligence (AI)-driven analysis has been reported to provide deeper insights than traditional statistical methods alone, demonstrating robust performance in disease classification, prediction, and identifying critical factors through the analysis of medical datasets [13,14]. Several studies have shown that models trained with various machine learning (ML) algorithms achieved high accuracy in predicting diseases like Parkinson’s disease, diabetes, and cardiac diseases [15,16,17]. Specifically, a study targeting Parkinson’s disease prediction utilized polygenic risk scores, clinical assessments, hospital admission examinations, and demographic data; both logistic regression and Extreme Gradient Boosting (XGBoost) demonstrated excellent performance, achieving an ROC AUC score of over 0.9 [15].

Other studies identified the key parameters that significantly impact disease classification, uncovering patterns that may not be visible through mere correlation or simple numerical comparisons [13,18,19,20]. A previous study on predicting breast cancer recurrence used the Shapley Additive Explanations (SHAP) method, an explainable AI approach based on game theory, analyzing the correlation between the patient’s prognosis and the features of high SHAP values, including CA125 [18].

AI has also been successfully integrated into studies involving patients with PTC. One study developed a recurrence prediction model using five algorithms, including Support Vector Machine (SVM) and Extreme Gradient Boosting (XGBoost), based on demographic data, lymph node-related variables, and metabolic and inflammatory markers [21]. In previous studies incorporating AI for the diagnosis of patients with PTC, various data types were tested for model training, including DNA methylation markers, nodule characteristics, autophagy-related genes from RNA sequencing data, and hub genes [22,23,24]. However, to the best of our knowledge, studies applying AI to discriminate patients with PTC based on serum multiple cytokine data have not been performed.

Therefore, we aimed to measure 46 cytokines simultaneously in both patients with PTC and healthy individuals, comparing the values of each cytokine between the two groups. Furthermore, we performed AI-driven analysis using various algorithms, evaluating their performance, and identifying the key parameters that classify disease and control groups.

## 2. Materials and Methods

### 2.1. Sample Collection and Preparation

Samples were remnant serum specimens after routine laboratory tests, which were archived in the Korea Institute of Radiological & Medical Sciences (KIRAMS) Radiation Biobank (KRB). The archived volume of samples was 0.3 mL and was stored in the KRB below −60 °C (Ultra-low temperature freezer MDF-U700VX-PK, Panasonic, Tokyo, Japan). We used distributed normal samples (*n* = 63) and PTC samples (*n* = 63) for quantification of targeted cytokines from KRB. Normal samples were residual plasma from health check-up patients who had no abnormalities in their examination results and were free of underlying diseases such as diabetes and hypertension. Distributed samples were enrolled from September 2019 to September 2023. The protocol was approved by the Institutional Review Board (IRB) of KIRAMS (KIRAMS-2023-10-004). The study was carried out in accordance with the Declaration of Helsinki [25]. The requirement for informed consent was waived because we used remnant specimens.

### 2.2. Clinical Data Collection

In both the PTC samples (*n* = 63) and normal samples (*n* = 63), the following laboratory findings were identified: hemoglobin, white blood cell (WBC) count, platelet count, neutrophil %, absolute neutrophil count, lymphocyte %, erythrocyte sedimentation rate (ESR), C-reactive protein (CRP), protein, albumin, total bilirubin, direct bilirubin, aspartate transaminase (AST), alanine aminotransferase (ALT), alkaline phosphatase (ALP), γ-glutamyl transferase (GGT), blood urea nitrogen (BUN), creatinine, uric acid, lactate dehydrogenase (LD), creatine kinase (CK), glucose, amylase, high-density lipoprotein (HDL)-cholesterol, low-density lipoprotein (LDL)-cholesterol, triglyceride, rheumatoid factor (RF), venereal diseases research laboratory (VDRL) test, and human immunodeficiency virus (HIV) test.

### 2.3. Multiple Cytokine Luminex Assay

The simultaneous quantification of cytokine levels in serum was performed using a commercially available Luminex assay Human XL Cytokine Panel (R&D systems, Minneapolis, MN, USA), according to the manufacturer’s protocol. The catalog number and company of all reagents is the same [LKTM014B and R&D Systems (Minneapolis, MN, USA)]. The measured cytokines were as follows: cluster of differentiation (CD) 40 Ligand, epidermal growth factor (EGF), eotaxin, fibroblast growth factor (FGF) basic, FMS-like tyrosine kinase (Flt)-3 Ligand, granulocyte colony-stimulating factor (G-CSF), granulocyte-macrophage (GM)-CSF, Granzyme B, chemokine (C-X-C motif) ligand (CXCL)1/GRO α, CXCL2/GRO β, interferon (IFN)-α2, IFN-β, IFN-γ, interleukin (IL)-1 α, IL-1 β, IL-1 receptor antagonist (IL-1ra), IL-2, IL-3, IL-4, IL-5, IL-6, IL-7, IL-8, IL-9, IL-10, IL-12 p70, IL-13, IL-15, IL-17A, IL-17E, IL-33, CXCL10/interferon-γ-induced protein (IP)-10, chemokine (C-C motif) ligand (CCL)2/monocyte chemoattractant protein (MCP)-1, CCL3/macrophage inflammatory protein (MIP)-1 α, CCL4/MIP-1β, CCL20/MIP-3α, CCL19/MIP-3β, platelet-derived growth factor (PDGF)-AA, PDGF-AB/BB, programmed death ligand 1 (PD-L1)/B7-H1, CCL5/regulated on activation, normal T cell expressed and secreted (RANTES), transforming growth factor (TGF)-α, tumor necrosis factor (TNF)-α, TNF-β, tumor necrosis factor-related apoptosis-inducing ligand (TRAIL), vascular endothelial growth factor (VEGF). The freeze–thaw cycle of samples was not repeated, while the Luminex assay was completed within 1 week after all the specimens were melted. When the signal is measured within the well, the system ensures that at least 50 beads per target protein are read, and it calculates the average fluorescence value of the phycoerythrin signal attached to these beads. Thus, the final result for each target protein in every well is derived from the signal detected on a minimum of 50 beads. The readout of cytokine fluorescence intensities was analyzed with the standard version of the Bio-Plex Manager software (version 6.1; Bio-rad, Hercules, CA, USA). A five-parameter logistic regression model was used to create standard curves and to calculate the marker concentration of each sample, expressed as pg/mL. The background of each analyte represents the fluorescence value of the well containing only the buffer used to dilute the sample and standard protein, serving as a negative control. Values of cytokine fluorescence intensities below the background were set to missing. The lower limit of quantification (LLOQ) was set at the lowest standard concentration for each cytokine, as presented in Appendix A. For those samples whose cytokine concentration was detected below the LLOQ, the value was set at LLOQ for the statistical analysis. All Luminex measurements were performed on the same day. To minimize analytical variability, all samples were analyzed as single samples in a one-plate-run modus. An external pool control from healthy donors ran in duplicate to calculate the coefficient of variance (CV). The median intra-assay CV for the pooled replicate sample was 2.8% across all analytes. The intra-assay CV for each analyte is presented in Appendix A. The median inter-assay CV for the pooled replicate sample was 8.5% across all analytes. The intra-assay CV for each analyte is presented in Appendix A. According to the standard curve, the assay working range of each cytokine is presented in Appendix A.

### 2.4. Data Processing and Statistical Analyses

Among the 46 cytokines and hematologic parameters, only those presenting numerical data were included in the analysis. Missing values were categorized into two scenarios: the first involved measurements that were out of range, and the second where data were absent. Parameters with more than 5% missing data among the 126 patients were excluded from the analysis [26,27]. For parameters with missing values below this threshold, the first scenario involved replacing values below the measurable range with the observed minimum value from the patient group [28] and those above the range with the maximum value [29]. In the second scenario, the mean value was utilized in place of missing data [30].

The statistical analysis was carried out using SPSS v27.0 (IBM, Armonk, NY, USA). The quantitative variables were presented as “median value [Quartile (Q)1, Q3]”, but as for the model performance of classification algorithms, measurements were presented as mean and 95% confidence interval. The Kolmogorov–Smirnov test was performed to examine whether those variables followed the normal distribution. To compare continuous measures between the PTC group and control group, Student’s independent samples *t*-test and corresponding Mann–Whitney U test were used for parametric and nonparametric analyses, respectively. All tests were conducted at the significance level *p* = 0.05.

### 2.5. Machine Learning Algorithms

In this study, we utilized five classification algorithms: K-nearest neighbor, Naïve Bayes classifier, logistic regression, Support Vector Machine (SVM), and Extreme Gradient Boosting (XGBoost). Among these, XGBoost showed the most superior performance across various fields [13,17,23]. Nested cross-validation (nCV) was implemented for the analysis. In the N × M nCV format, the dataset is divided into N outer folds, with each fold reserved for testing and the remaining N-1 folds combined and divided into M inner folds for hyperparameter tuning [31]. This strategy prevents the overlap of data used for hyperparameter tuning and model evaluation, ensuring that all data can act as test data and lowering the risk of overfitting due to arbitrary test splits [31,32]. Furthermore, this method reduces bias during hyperparameter tuning and yields a more precise evaluation of the model’s generalization ability compared to single cross-validation, thus supporting its application in our study [31,32].

Due to the large variance observed in the 5-outer fold caused by the small sample size in the test fold, the total dataset was divided into a 4-outer fold for analysis, the total dataset (*n* = 126) was divided into four outer folds for analysis (Figure 1). After merging the non-test folds for training, the training fold was further split into five inner folds to determine the hyperparameters through a cross-validation process. Hyperparameter tuning was conducted using the Grid Search method, which explores all possible combinations within the hyperparameter grid to identify the optimal results. The candidate values for the hyperparameter grid were set based on previous studies [13,20], while the tuned hyperparameters are presented as follows. For the k-NN model, the values of k and the distance metric were optimized, whereas for the SVM model, the regularization parameter and kernel coefficient were selected. In the case of XGBoost, the learning rate, maximum depth of each tree (max_depth), minimum loss reduction (gamma), total number of trees added to the model (n_estimators), and the proportion of samples used for fitting the individual trees (subsample) were determined. For the logistic regression model, regularization type and regularization strength (C) were selected. For the Naïve Bayes classifier, a smoothing parameter was provided for tuning. Classification performance is assessed by accuracy, precision, sensitivity (recall), specificity, f1-score, and area under the curve-receiver operator characteristic (AUC-ROC) score. To minimize the risk of chance findings in data splitting, we repeated this process five times, using the average of 20 scores to evaluate each algorithm’s performance. To achieve clearer explanations of decisions made by artificial intelligence models, we applied the SHAP method, which uses game theory principle to provide parameter’s importance for prediction and how it influences the likelihood of prediction outcomes [24]. SHAP values quantify how much each feature contributes to a specific prediction. In this study, we calculated SHAP values for each feature in the best-performing model for every fold and computed their averages. Utilizing the SHAP summary plot and beeswarm plot, we clarified the impact and directionality of contributions from various parameters on the classification of patients with PTC. Tuning and training processes including grid search, cross-validation and training algorithms, were performed using the XGBoost package (version 2.1.1) and the Python scikit-learn package (version 1.4.2). SHAP analysis was conducted using the Python SHAP library (version 0.40.0), while the graphics were created using the Python Matplotlib library (version 3.8.4) and the Python Seaborn library (version 0.13.2).

## 3. Results

### 3.1. Patient Characteristics

The following parameters with missing values over 5% were excluded: ESR, CRP, protein, albumin, total bilirubin, direct bilirubin, GGT, BUN, uric acid, LD, CK, amylase, HDL-cholesterol, LDL-cholesterol, triglyceride, and RF. As for the 63 patients and 63 healthy controls, the demographic features and laboratory parameters of the PTC and control group are shown in Table 1. Age was significantly higher in the PTC group than in the control group (*p* = 0.002, Mann–Whitney U test). The gender ratio was the same in both groups: 30 males and 33 females. While there were features that showed differences between the two groups, values of all laboratory parameters remained within normal ranges.

### 3.2. Comparison of Serum Cytokines Between PTC and Control Group

Thirteen cytokines with missing values exceeding 5% of patients were excluded from the analysis: fibroblast growth factor (FGF) basic, granulocyte colony-stimulating factor (G-CSF), interferon (IFN)-α 2, IFN-β, interleukin (IL)-2, IL-3, IL-4, IL-5, IL-9, IL-17A, IL-17E, chemokine (C-C motif) ligand (CCL)2/macrophage inflammatory protein (MIP)-1 α, and tumor necrosis factor-related apoptosis-inducing ligand (TRAIL). The Kolmogorov–Smirnov test was conducted to determine if the variables followed a normal distribution. Seven cytokines—CCL11/Eotaxin, IL-7, IL-15, CCL2/MCP-1, PD-L1/B7-H1, CCL5/RANTES, and TNF-β—were found to meet this criterion. To compare continuous measures between two groups, Student’s *t*-test was utilized for these seven cytokines, whereas the Mann–Whitney U test was applied for the others. The results are presented in Table 2.

The difference between two groups was significant for 15 cytokines (Table 2), and these significant cytokines are represented in Figure 2.

### 3.3. Comparative Analysis of Algorithms and Feature Importance Using Explainable Artificial Intelligence

Performance scores were obtained through five repetitions of nCV. A five-fold split was first performed for the outer fold, with a test size of 25–26 set for a total of 126 samples. Upon calculating the coefficient of variation for the 25 performance scores across five algorithms and five folds each, an average value of 10.0% was obtained. However, certain metrics, such as the k-NN Sensitivity score and the NB Sensitivity score, exceeded 20% or approached this threshold, indicating high variance (Appendix A). To mitigate this issue, the outer fold was adjusted to a four-fold split, increasing the test fold size. As a result, the coefficient of variation was reduced to an average of 7.4%, with a maximum of 18.8%.

Table 3 summarizes the performance of classification algorithms based on accuracy, precision, sensitivity (recall), F1-score, specificity, and ROC-AUC score. According to the mean score of each algorithm, XGBoost demonstrated superiority over the other algorithms across all five metrics. Among traditional algorithms, logistic regression showed the highest performance.

Figure 3 compares the empirical distributions of the five algorithms. Based on the median, as with the mean, XGBoost outperformed all other algorithms across all six metrics, followed by logistic regression. However, in terms of sensitivity, XGBoost showed a marginal difference compared to logistic regression and SVM, whereas its median ROC AUC score was similar to that of logistic regression.

SHAP analysis was conducted for each XGBoost model, and the mean SHAP values for each feature were calculated across a total of 20 models, as presented in Figure 4. EGF exhibited the highest SHAP value, followed by IL-10, CD40 Ligand, and IL-1β. Notably, lower levels of EGF, CD40 Ligand, and IL-1β were correlated with higher SHAP values, whereas higher concentrations of IL-10 were associated with higher SHAP values. Consequently, EGF was identified as the most influential feature in distinguishing patients with PTC, with IL-10 following.

### 3.4. Classification Performance of XGBoost According to the Increase in the Number of Features (In Descending Order of SHAP Values)

Figure 5 illustrates the classification performance of XGBoost when the features were selected according to their SHAP values in descending order, as presented in Figure 4. Similarly to the previous methods, 20 performance scores were obtained by repeating the 4 × 5-fold cross-validation five times and utilizing their average. Notably, performance evaluation scores exceeding 0.8 across five metrics in Figure 5 were achieved using only the two features, EGF and IL-10. And with the two features, specificity showed an average value of 0.798.

### 3.5. Concentration Distribution of CXCL10/IP-10 in Patients with PTC and the Normal Cohort

Figure 6 depicts the concentration distribution of CXCL10/IP-10 in patients with PTC and the normal cohort, indicating a significant overlap.

## 4. Discussion

PTCs can be treated effectively if detected at an early stage. However, most diagnoses are made through invasive biopsies, which typically detect the condition at more advanced stages. Therefore, our study aimed to measure multiple cytokines for the early diagnosis of PTC in clinical settings, performing various AI algorithms based on the data and exploring the key parameters that classify disease and control groups.

For all data (63 patients with PTC and 63 controls), a hierarchical sampling method was used during the cross-validation process to ensure a balanced dataset for both training and testing. As a result, the evaluation metrics—accuracy, precision, F1-score, sensitivity and specificity—were meaningful indicators of performance, with XGBoost showing the highest performance. A previous study that compared the predictive performance of algorithms including XGBoost, SVM, k-NN, and LR for hematoma expansion prediction on a balanced dataset found that XGBoost outperformed the other models with accuracy, precision, recall (sensitivity), and an F1-score of 0.82, consistent with our findings [33]. However, in contrast to the current study, LR in the previous study exhibited lower performance than SVM and k-NN. This discrepancy can be attributed to the use of penalized LR in the current study to prevent overfitting, unlike the standard logistic regression used previously [34]. Indeed, another study comparing the predictive performance of several classification algorithms, including k-NN, LR, SVM, and XGBoost, for Parkinson’s disease risk prediction found that penalized logistic regression and XGBoost yielded the highest performance, with LR even surpassing XGBoost in terms of AUC [15].

Meanwhile, the lower performance of the k-NN algorithm and Naïve Bayes compared to other algorithms is attributed to the curse of dimensionality. In the case of XGBoost, it employs a boosting algorithm, which assigns higher weights to misclassified data points, enabling it to identify important patterns even in high-dimensional data [35]. Additionally, hyperparameters such as tree depth and the number of trees are included, and the use of random sampling techniques helps prevent overfitting, making it robust against the curse of dimensionality. For logistic regression, the inclusion of an L2 regularization term helps mitigate overfitting, while in the case of SVM, the hyperparameters gamma and C can be adjusted to address the issue to some extent [36]. On the other hand, the Naïve Bayes classifier, due to its assumption of feature independence, and the k-NN algorithm, which becomes more susceptible to the curse of dimensionality as the number of dimensions increases (due to the growing number of distance calculations), are more vulnerable to this issue.

To perform dimensionality reduction, PCA analysis was conducted, revealing that the cumulative explained variance reached 0.9 when the number of features was reduced to 22 (Appendix A). Using the dataset with 22 components, k-NN and Naïve Bayes were evaluated through five iterations of 4 × 5 nested cross-validation, similar to the original setup. The results showed notable performance improvement, especially for k-NN, compared to the pre-reduction phase (Appendix A). However, the improvement was not significant enough to overturn the overall ranking, except for NB and SVM. In fact, projecting the existing features onto a lower-dimensional space or forming new principal components for training the AI would make it difficult to assess the extent to which the original features influence the classifier, as it would obscure their importance when using the SHAP method. For the reasons mentioned, we did not preemptively perform dimensionality reduction during AI training.

Since XGBoost outperformed other models, SHAP analysis was performed on 20 XGBoost models, and the means of SHAP values for each feature were calculated. Among the features, EGF was identified as the most influential factor in the classification by XGBoost. As depicted in Figure 4, the SHAP value for EGF was higher than for other factors. A lower EGF level correlates with a higher probability of being classified as a patient with PTC, consistent with the EGF boxplot shown in Figure 2. Supporting this, one study performing cytokine multiplex analysis indicated that the PTC group had lower EGF serum levels than the control group [9]. Conversely, another study found that patients with PTC (*n* = 48) had higher EGF serum levels than healthy participants (*n* = 20) [37], with patients at stages T2, T3, and T4 displaying significantly higher EGF serum levels, while those at stage T1 had significantly lower levels than controls. Although pathological staging data were not collected for patients with PTC in our study, it is possible that our participants had lower pathological stages.

The subsequent most influential factor identified was IL-10. Higher levels of IL-10 are associated with an increased likelihood of being classified as a patient with PTC (Figure 4). Consistently, Figure 2 shows that IL-10 serum levels are elevated in patients with PTC compared to controls. Unlike EGF, IL-10 consistently demonstrated higher levels in patients with PTC than in healthy controls in prior studies [5,38]. Similarly, the top five features with high SHAP values, with the exception of CXCL10/IP-10, displayed significant differences between patients with PTC and healthy controls (Figure 2 and Figure 4, and Table 2). This indicates that the XGBoost algorithm effectively incorporated these differences in forming its decision trees. The performance, as the number of features with high SHAP values increased, was assessed.

Figure 5 emphasizes that the serum levels of cytokines, particularly EGF and IL-10 (having the highest SHAP values), can provide substantial assistance in determining whether a patient has PTC. Meanwhile, CXCL10/IP-10, which exhibited no significant difference between patients with PTC and normal controls (Table 2 and Figure 6), contributed to an increase in five performance metrics, excluding precision (Figure 5). Figure 6 depicts the concentration distribution of CXCL10/IP-10 in patients with PTC and the normal cohort, indicating a significant overlap (distributions of other features with high SHAP values are provided in Appendix A). Although simply setting a threshold of CXCL10/IP-10 to distinguish patients with PTC may not be effective for identification (Figure 6), within the XGBoost model, CXCL10/IP-10 plays a crucial role in enhancing performance. While patients with PTC had a higher average concentration of CXCL10/IP-10 compared to the control (Table 2), the model suggests that lower levels of CXCL10/IP-10 increase the likelihood of being a patient with PTC. This indicates that the XGBoost model provides a more comprehensive analysis by integrating the distribution with multiple trees rather than solely making comparisons. Indeed, a previous study that identified overlapping data, namely creatine, as a significant parameter in XGBoost for classification results, suggested that machine learning offers a more nuanced analysis than traditional statistical methods [13]. Thus, even without significant differences between the groups, screening for CXCL10/IP-10 can enhance the performance of machine learning models.

Demographic and laboratory data showed no significant differences except in age, ALT, and glucose levels (Table 1). However, ALT and glucose levels remained within the normal range for both the PTC and control groups, indicating that they likely did not affect the differences in cytokine levels. The measurement results revealed significant differences between the PTC and control groups for CD40 Ligand (CD40L), EGF, GM-CSF, IFN-γ, IL-1β, IL-1ra, IL-7, IL-10, IL-12p70, IL-15, CCL20/MIP-3α, PDGF-AA, CCL5/RANTES, TGF-α, and TNF-α (Table 2, Figure 2). The other cytokines showed no significant differences between the two groups (Table 2).

CD40 is a cell surface receptor belonging to the tumor necrosis factor receptor superfamily [39]. Its interaction with CD40L, which is typically expressed on activated T cells, initiates several immune-related processes, such as B cell proliferation, antibody class switching, and T cell activation [39]. Regarding the expression of CD40 and CD40L in thyroid tissues, a previous study immunohistochemically stained those from 36 PTC and four follicular TC (FTC), demonstrating that most thyroid tumors expressed CD40 and CD40L. In that study, patients ≤21 years old with intense CD40L expression presented more multifocal, aggressive, and recurrent tumors [40]. However, in our study, although the PTC group had lower CD40L levels than the control group, the specimen analyzed was serum, not thyroid tissues. Thus, while thyroid tissues of patients with PTC in our study could potentially exhibit CD40L expression, further research is required to verify CD40L expression in thyroid tissues.

TGF-α, a small polypeptide, competes with EGF for binding to the shared receptor, EGFR [41]. EGFR, a transmembrane receptor, initiates signaling pathways such as RET/PTC rearrangement and MMP-2/gelatinase A upon activation by ligands like TGF-α, which promote cell proliferation, differentiation, migration, and apoptosis [41,42,43]. Regarding TGF-α expression in thyroid tissues, a previous study utilizing real-time quantitative polymerase chain reaction and immunohistochemistry examined 71 PTCs, 68 paired non-cancer thyroid tissues adjacent to the PTC, and 20 benign thyroid lesions. This study found that PTCs expressed higher levels of TGF-α compared to benign thyroid lesions [41]. In our study, the PTC group exhibited lower TGF-α levels than the control group, but the specimens were serum, not thyroid tissues. Consequently, future research is essential to clarify the differences in expression between thyroid tissue and serum.

GM-CSF plays a crucial role in enhancing myeloid cell survival and activation, which may influence immune surveillance and the aggressiveness of cancer cells [44]. Neutrophil survival facilitated by GM-CSF leads to the secretion of various pro-inflammatory and angiogenic factors, such as IL-8, VEGF-A, TNF-α, and MMP-9, all contributing to tumor progression and angiogenesis [45]. Consequently, GM-CSF is a vital mediator that enables thyroid cancer cells to foster a conducive environment for tumor progression by influencing neutrophil behavior [45]. Consistent with these findings, our study demonstrated that the PTC group had significantly higher serum GM-CSF levels compared to the control group.

IFN-γ, a cytokine secreted by immune cells, including T and NK cells, plays a crucial role in modulating the immune response against tumors. In PTC, IFN-γ stimulates the release of chemokines, notably CXCL9 and CXCL11, which exert antiproliferative effects on PTC cells and restrict cancer cell migration [46,47]. Thus, elevated IFN-γ levels in patients with PTC in our study likely signify an active immune response targeting tumor cells. Although IFN-γ typically supports anti-tumor immunity, the inflammatory environment prevalent in many cancers, such as PTC, can also promote IFN-γ production by tumor cells and adjacent stromal and immune cells [46,47].

IL-10 is an anti-inflammatory cytokine that functions as a key mediator of anti-cancer strategies by activating CD8+ T cells [48]. Activated CD8+ T cells attack MHC-positive cancer cells [48] and PTCs often reduce MHC class I expression to evade them [49]. Therefore, increased IL-10 can be inferred as a result of immune evasion in PTC [5,48]. One previous study reported that 20 thyroid cancer patients had significantly higher IL-10 levels than 50 healthy individuals [5]. Another study found that 16 patients with PTC exhibited higher IL-10 serum levels than 24 healthy participants, although the difference was not statistically significant [50]. Similarly, a study involving 186 patients with PTC showed increased IL-10 serum levels compared to 100 healthy participants, with no significant difference [10]. However, our study indicates that the PTC group had significantly higher IL-10 levels than the control group, corroborating the pathogenesis outlined in the literature [48,49].

A prior study involving 20 thyroid cancer patients and 50 healthy controls found that the thyroid cancer patient group had lower IL-1β levels than the control group, but the difference was not statistically significant [5]. While our study focused solely on patients with PTC within the thyroid cancer category, the PTC group exhibited significantly lower IL-1β levels than the control group.

Regarding IL-1ra serum levels, a previous study included 21 PTC, 8 FTC, 12 medullary TC, 11 anaplastic TC patients, and 27 healthy volunteers, finding that IL-1ra was significantly higher only in anaplastic and follicular TC patients compared to the control group [51]. As for IL-1ra tissue expression, another study reported higher levels in PTC tissues than in normal tissues [52]. Our study found a significant increase in IL-1ra serum levels in the PTC group compared to the control group, necessitating further investigation with more specimens to resolve the discrepancy to the previous serum IL-1ra study and identify IL-1ra tissue expression in PTC group.

A previous study utilizing multiplex Luminex assay on 23 thyroid cancer patients, including patients with PTC, and 23 healthy participants, indicated that the serum IL-15 levels in the thyroid cancer group were not significantly different from those in the control group; this study, however, did not examine IL-15 differences specifically between patients with PTC and the control group [9]. In contrast, our study found elevated IL-15 levels in PTC cases relative to the control group.

IL-7 and IL-12p70 demonstrated significant differences between patients with PTC and the control group in our research, but this cytokine has not been extensively studied previously, highlighting the need for further research.

A prior study using immunohistochemical staining on thyroid tissues revealed that PTCs with nodal metastases exhibited higher PDGF-AA expression compared to benign thyroid adenoma and PTCs without nodal metastases, though it did not assess PDGF-AA serum levels [53]. Conversely, our study found reduced PDGF-AA levels in the patient group with PTC compared to healthy controls. Future research must explore and clarify the variance in PDGF-AA levels between serum and thyroid tissues in patients with PTC.

CCL20/MIP-3α acts as a ligand for CCR6 (C-C chemokine receptor type 6) [8]. CCR6 has been found to be significantly upregulated in thyroid cancer cells compared to normal thyroid epithelial cells [54]. The CCL20/CCR6 axis contributes to the activation of NF-kB and secretion of MMP-3, which promote cancer cell invasion and migration [54]. Our findings showing higher serum levels of CCL20/MIP-3α in the PTC group than in the control group corroborate these earlier studies.

The elevation of CCL5/RANTES levels in patients with PTC in our study may be attributed to several factors related to the tumor microenvironment and immune response dynamics. A previous study demonstrated that CCL5/RANTES could play a significant role in the development and progression of autoimmune thyroid disorders and that IL-1 and TNF-α induced strong RANTES secretion by thyroid-derived fibroblasts [55]. Consistent with this, our study showed that patients with PTC exhibited elevated levels of TNF-α and CCL5/RANTES compared to the control group. Conversely, one previous study reported significantly lower levels of CCL5/RANTES in 23 thyroid cancer patients (18 patients with PTC) compared to 23 healthy participants, while our study included more participants (63 patients with PTC and 63 healthy participants) [9]. Moreover, previous research has acknowledged the possibility of undiagnosed Graves’ disease cases within the control group, which could potentially confound the interpretation of results [9]. TNF-α stimulation was reported to increase CCR6+ cells in thyroid tumor lines TPC-1 and BCPAP [56]. As TNF-α enhances CCR6 expression, CCR6+ cells interact with CCL20, facilitating migration in cancer cells but not affecting normal thyroid cells [56]. This observation is corroborated by our study, which found increases in both TNF-α and CCL20/MIP-3α levels in the PTC group compared to the control group.

Our study had several limitations. First, the evaluation was conducted using a single measurement rather than duplicate measurements. But internal controls and a standard curve were utilized to minimize potential variations during the experimental process. Additionally, a validated protocol (standard operating procedure) was used, confirmed through standard materials and repeated experiments (Luminex performance assay; Human XL Cytokine Fixed Panel; catalog number LKTM014B; R&D Systems). Furthermore, to reduce variability between samples, the analysis was conducted under identical conditions. Second, the potential impact of missing values on the results cannot be overlooked. To address this issue, only parameters with less than 5% missing data were considered, and missing values were managed using methods established in previous studies, thereby enhancing the reliability of our findings [26,27,28,29,30]. Lastly, the patient and control groups were matched in size, each comprising 63 individuals; however, demographic data (Table 1) indicated a potential age-related bias within the dataset. Notably, the highest proportion of patients was observed in the 60s and 70s. Consequently, age was excluded as a variable in AI model training to mitigate the potential bias, as older patients might be preferentially diagnosed with PTC regardless of the cytokine levels.

The findings of this study are useful in identifying differences in cytokine profiles between PTC patients and healthy controls and suggest the potential for developing an automated PTC prediction model using cytokine data. In particular, EGF and IL-10 may serve as key biomarkers for PTC diagnosis. Future multi-center studies with larger sample sizes are needed to validate these findings and to develop auxiliary diagnostic tools utilizing serum cytokine panels.

Additionally, since this study was conducted in a specific population (PTC patients vs. healthy individuals), the methodology used in this study could also be applied to other thyroid cancers, such as follicular, anaplastic, and medullary thyroid carcinoma, as well as benign thyroid nodules and Hashimoto’s thyroiditis, to identify serum cytokines uniquely associated with each condition. Through this approach, it may be possible to develop a serum cytokine panel test that can distinguish between different types of thyroid cancer, benign nodules, and other thyroid disorders. Then, thyroid cancer patients undergoing blood tests (such as TSH, T3, free T4, anti-thyroglobulin, calcitonin, and thyroglobulin) for diagnosis or follow-up examinations could undergo the developed cytokine panel test, which could serve as a useful supplementary tool for diagnosis and recurrence monitoring. In addition, although this study measured 46 cytokines using the Luminex assay, further validation is necessary to confirm whether the same patterns are observed in cytokine data obtained through other methods, such as enzyme-linked immunosorbent assay, meso scale discovery, and flow cytometry.

In conclusion, our study analyzed 46 cytokines in the serum of 63 patients with PTC and 63 healthy individuals. Among these, 15 cytokines exhibited statistically significant differences between the two groups. Of these 15 cytokines, only 5 were significantly lower in the PTC group compared to the healthy individuals, whereas the remaining 10 were significantly higher. Additionally, we trained various classification algorithms using these data, of which XGBoost demonstrated the best performance. SHAP analysis was performed on the XGBoost model to identify the key cytokines that significantly contribute to distinguishing patients with PTC. Notably, EGF and IL-10 are believed to be associated with the pathogenesis of PTC, and further research is required to elucidate their roles.

## Figures and Tables

**Figure 1 diagnostics-15-00791-f001:**
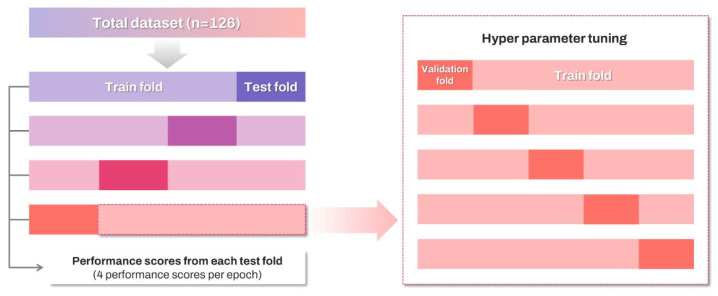
The total dataset was split into four outer folds for training and evaluation, while the training fold was split into five inner folds for hyperparameter tuning.

**Figure 2 diagnostics-15-00791-f002:**
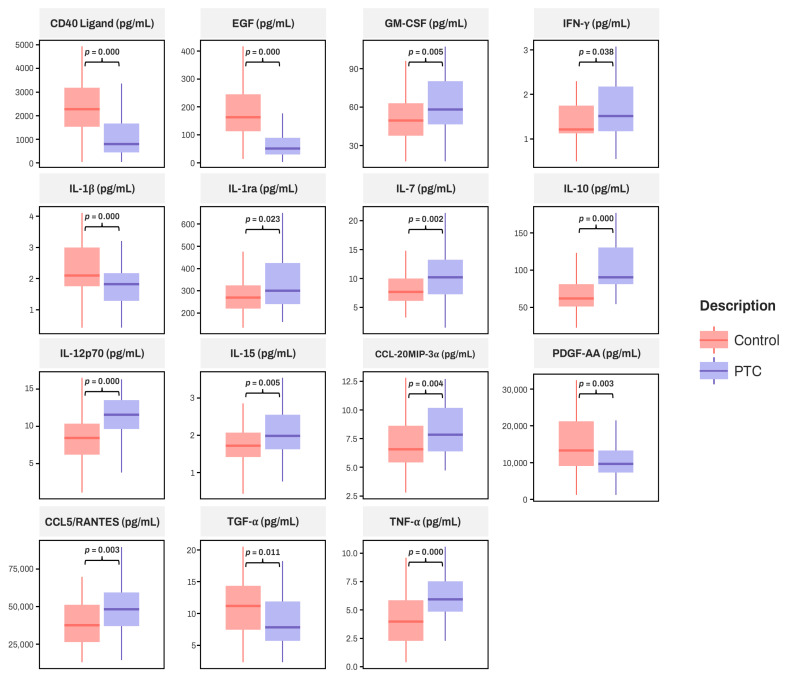
Comparison of 15 cytokines between the PTC group and the control group, with significant differences noted between the two groups. *p*-values are displayed above the boxplot.

**Figure 3 diagnostics-15-00791-f003:**
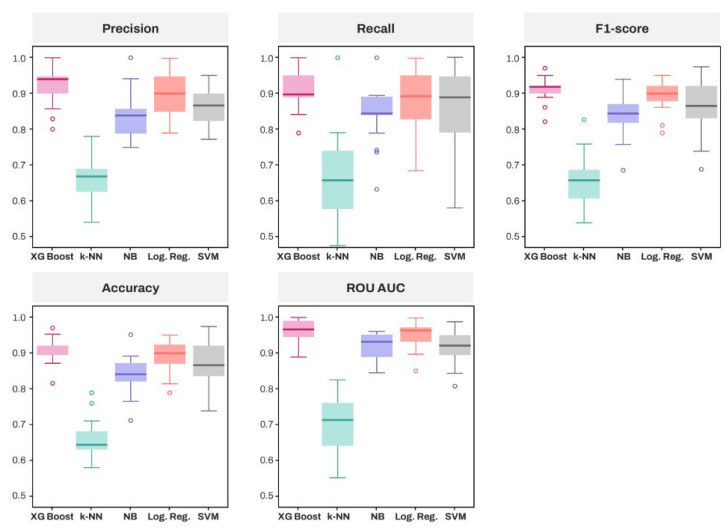
Empirical distribution of model performance of five classification algorithms. Five performance measurements were calculated; Precision, Recall (Sensitivity), F1-score, Accu-racy, ROC-AUC score.

**Figure 4 diagnostics-15-00791-f004:**
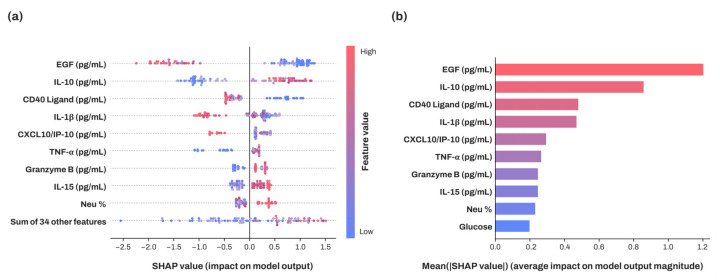
SHAP analysis for XGBoost classifier. (**a**) Beeswarm plot illustrating each feature’s effect on SHAP value (model output). (**b**) Absolute SHAP values shown as bar plot.

**Figure 5 diagnostics-15-00791-f005:**
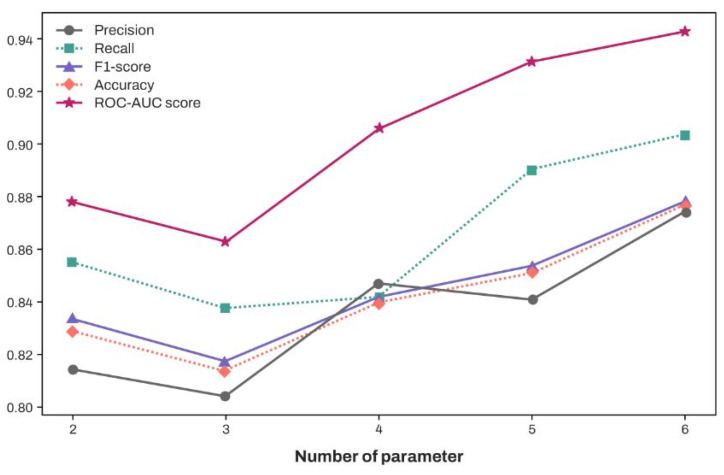
Performance of XGBoost using data with limited features exhibiting high SHAP values, starting with EGF and IL-10 and progressively incorporating additional features based on their SHAP values: CD40 Ligand, IL-1β, CXCL10/IP-10, TNF-α.

**Figure 6 diagnostics-15-00791-f006:**
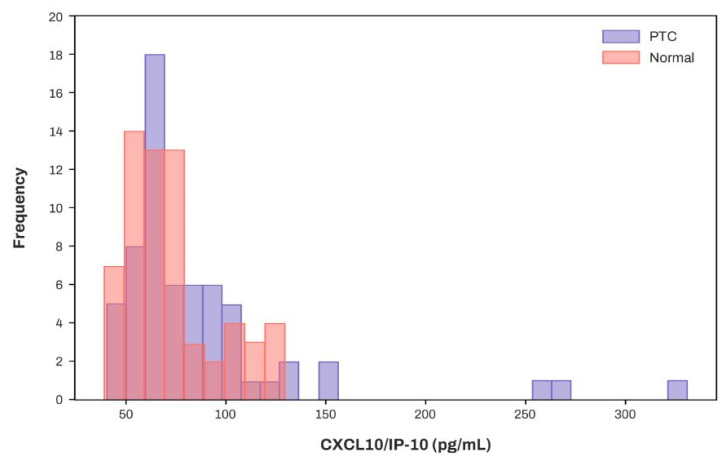
Distribution of CXCL10/IP-10 in PTC and control groups. Abbreviations: CXCL—C-X-C motif chemokine ligand; CXCL10/IP-10—CXCL10/interferon γ-induced protein 10; PTC—Papillary thyroid carcinoma.

**Table 1 diagnostics-15-00791-t001:** Demographic features and laboratory parameters of each PTC and control group.

Group	Control [*n* = 63]	PTC [*n* = 63]	*p*-Value
Age [years]	40 (36, 47)	47 (38, 59)	0.002
Gender [Male/Female]	30:33	30:33	-
Hb [g/dL]	13.8 (13.0, 14.9)	14.0 (13.0, 15.3)	0.368 *
WBC [10^3^/μL]	5.70 (4.92, 6.62)	5.62 (4.78, 6.60)	0.982 *
PLT [10^3^/μL]	236.0 (210.0, 278.5)	246.0 (201.0, 267.0)	0.685 *
Neu [%]	55.80 (48.50, 59.65)	55.90 (51.10, 62.90)	0.225 *
ANC [10^3^/μL]	3.030 (2.505, 3.840)	3.140 (2.420, 4.160)	0.513 *
Lym [%]	35.40 (29.75, 39.25)	32.40 (27.70, 39.40)	0.311 *
AST [U/L]	19.0 (17.0, 23.0)	19.0 (16.0, 24.0)	0.391
ALT [U/L]	15.0 (12.0, 20.0)	20.0 (13.0, 28.0)	0.024
Cr [mg/dL]	0.80 (0.60, 0.90)	0.80 (0.60, 0.90)	0.878
Glucose [mg/dL]	88.00 (83.00, 94.00)	97.50 (90.75, 115.75)	<0.001

Quantitative data are presented as median (Q1, Q3). *, Student’s *t*-test; otherwise, a nonparametric Mann–Whitney U test was performed for comparison. Abbreviations: ALT—alanine aminotransferase; ANC-absolute neutrophil count; AST—aspartate aminotransferase; Cr—creatinine; Hb—hemoglobin; Lym—lymphocyte; Neu—neutrophil; PLT—platelet; PTC—papillary thyroid cancer; WBC—white blood cell.

**Table 2 diagnostics-15-00791-t002:** Comparison of serum cytokines in PTC and control group.

Group	Control [*n* = 63]	PTC [*n* = 63]	*p*-Value
CD40 Ligand [pg/mL]	2252.26 (1492.46, 3175.59)	813.54 (448.03, 1680.63)	<0.001 *
EGF [pg/mL]	163.42 (109.89, 245.82)	54.12 (28.40, 88.56)	<0.001
CCL11/Eotaxin [pg/mL]	116.08 (95.34, 153.24)	125.46 (92.48, 161.13)	0.248 *
Flt-3 Ligand [pg/mL]	64.01 (55.81, 73.27)	66.63 (55.14, 74.75)	0.682
GM-CSF [pg/mL]	49.75 (36.00, 62.70)	58.61 (46.44, 80.85)	0.005
Granzyme B [pg/mL]	12.78 (10.72, 15.74)	13.35 (11.57, 16.34)	0.766
CXCL1/GRO α [pg/mL]	89.21 (73.14, 139.72)	89.50 (60.68, 119.09)	0.271
CXCL2/GRO β [pg/mL]	549.23 (412.86, 831.90)	491.82 (340.64, 712.72)	0.130
IFN-γ [pg/mL]	1.22 (1.13, 1.80)	1.51 (1.13, 2.20)	0.038
IL-1α [pg/mL]	7.11 (4.05, 8.99)	7.11 (5.30, 7.98)	0.809
IL-1β [pg/mL]	2.59 (1.67, 3.05)	1.82 (1.12, 2.17)	<0.001
IL-1ra [pg/mL]	274.31 (218.39, 326.87)	301.52 (238.06, 435.44)	0.023
IL-6 [pg/mL]	5.64 (4.72, 6.57)	5.64 (4.68, 7.57)	0.163
IL-7 [pg/mL]	7.58 (6.25, 9.83)	10.16 (7.31, 13.44)	0.002 *
IL-8/CXCL8 [pg/mL]	11.15 (6.22, 33.55)	8.91 (6.16, 16.14)	0.116
IL-10 [pg/mL]	62.37 (49.65, 80.68)	91.31 (80.68, 131.34)	<0.001
IL-12p70 [pg/mL]	8.39 (6.29, 10.45)	11.53 (9.17, 13.92)	<0.001
IL-13 [pg/mL]	16.87 (12.02, 21.35)	19.39 (12.19, 23.17)	0.812
IL-15 [pg/mL]	1.74 (1.41, 2.07)	2.00 (1.65, 2.60)	0.005 *
IL-33 [pg/mL]	16.87 (12.02, 21.35)	19.39 (12.19, 23.17)	0.291
CXCL10/IP-10 [pg/mL]	67.74 (57.32, 80.64)	69.82 (60.66, 96.84)	0.099
CCL2/MCP-1 [pg/mL]	211.39 (175.73, 258.09)	213.48 (171.11, 267.43)	0.606 *
CCL4/MIP-1β [pg/mL]	488.36 (411.46, 671.39)	453.88 (381.14, 583.87)	0.117
CCL20/MIP-3α [pg/mL]	6.55 (5.42, 8.95)	7.86 (6.27, 10.37)	0.004
CCL19/MIP-3β [pg/mL]	62.95 (51.38, 85.27)	73.35 (51.91, 91.16)	0.237
PDGF-AA [pg/mL]	13,543.28 (8996.78, 22,339.41)	9912.86 (7623.70, 13,417.70)	0.003
PDGF-AB/BB [pg/mL]	4112.81 (2906.69, 5404.23)	3944.59 (2985.52, 4990.56)	0.691
PD-L1/B7-H1 [pg/mL]	68.73 (57.61, 82.54)	65.32 (47.46, 74.65)	0.052 *
CCL5/RANTES [pg/mL]	37,894.82 (26,416.35, 52,197.53)	47,847.57 (36,220.00, 59,566.93)	0.003 *
TGF-α [pg/mL]	11.23 (7.53, 14.73)	7.82 (5.64, 12.09)	0.011
TNF-α [pg/mL]	3.96 (2.34, 5.83)	5.96 (4.93, 7.51)	<0.001
TNF-β [pg/mL]	2.74 (2.15, 3.43)	2.95 (2.12, 3.79)	0.249 *
VEGF [pg/mL]	173.19 (121.67, 216.58)	153.68 (109.91, 200.01)	0.336

Quantitative data are presented as median (Q1, Q3). *, Student’s *t*-test; otherwise, a nonparametric Mann–Whitney U test was performed for comparison. Abbreviations: CCL—C-C motif chemokine ligand; CD—cluster of differentiation; CXCL—C-X-C motif chemokine ligand; CXCL10/IP-10—CXCL10/interferon γ-induced protein 10; EGF—epidermal growth factor; Flt-3 Ligand—FMS-like tyrosine kinase (Flt)-3 Ligand; GM-CSF—granulocyte-macrophage colony-stimulating factor; IFN-γ—interferon-γ; IL—interleukin; MCP-1—monocyte chemoattractant protein 1; IP—interferon γ-induced protein; MIP—macrophage inflammatory protein; PDGF—platelet-derived growth factor; PD-L1—programmed death-ligand 1; PTC—papillary thyroid cancer; RANTES—regulated on activation, normal T cell expressed and secreted; TGF—transforming growth factor; TNF—tumor necrosis factor; VEGF—vascular endothelial growth factor.

**Table 3 diagnostics-15-00791-t003:** Model performance for classification algorithms.

Model	Accuracy	Precision	Sensitivity (Recall)	F1-Score	Specificity	ROC-AUC Score
XGBoost	0.913 (0.897, 0.928)	0.922 (0.896, 0.948)	0.913 (0.888, 0.938)	0.912 (0.897, 0.928)	0.920 (0.889, 0.950)	0.964 (0.950, 0.977)
k-NN	0.653 (0.625, 0.681)	0.658 (0.629, 0.686)	0.661 (0.604, 0.718)	0.652 (0.620, 0.684)	0.646 (0.592, 0.700)	0.703 (0.668, 0.738)
Log. Reg.	0.892 (0.870, 0.920)	0.900 (0.850, 0.946)	0.886 (0.829, 0.952)	0.890 (0.880, 0.920)	0.897 (0.862, 0.932)	0.950 (0.931, 0.971)
NB	0.841 (0.871, 0.913)	0.840 (0.870, 0.932)	0.847 (0.848, 0.925)	0.840 (0.868, 0.912)	0.835 (0.799, 0.872)	0.919 (0.933, 0.967)
SVM	0.866 (0.835, 0.896)	0.866 (0.841, 0.892)	0.882 (0.811, 0.918)	0.862 (0.828, 0.897)	0.867 (0.837, 0.896)	0.916 (0.893, 0.938)

Mean and 95% confidence interval for six measurements were presented for each model. Abbreviations: k-NN—k-nearest neighbor; Log. Reg.—Logistic Regression; NB—Naïve Bayes; SVM—Support Vector Machine; XGBoost—Extreme gradient boosting.

## Data Availability

The original contributions presented in this study are included in the article. Further inquiries can be directed to the corresponding author.

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
