# Peer review of "Comparison of 46 Cytokines in Peripheral Blood Between Patients with Papillary Thyroid Cancer and Healthy Individuals with AI-Driven Analysis to Distinguish Between the Two Groups"

_diagnostics, 2025, doi:10.3390/diagnostics15060791_

Round 1

Reviewer 1 Report

Comments and Suggestions for Authors

Peer-review

Comparison of 46 cytokines in peripheral blood between patients with papillary thyroid cancer and healthy individuals with AI-driven analysis to distinguish between the two groups

Diagnostics

Comment: It is a very well planned and described study. Congratulations!

Title

No comments

Abstract

No comments

Introduction

No comments

Methods

Recommendation: Please specify the software and packages used for graphics creation (e.g., Python libraries such as Matplotlib, Seaborn, or others) to enhance transparency and reproducibility.

Results

Recommendation: The presentation of the statistical analysis results appears redundant—for example, the details provided in Table 2, the accompanying text, and the figure seem to overlap. Please verify whether this repetition is necessary and consider streamlining the presentation to avoid excessive duplication of results.

Discussion

Recommendation: Please consider moving Figures 5 and 6 to the Results section to provide a clearer separation between data presentation and discussion, enhancing the manuscript's overall clarity and structure.

Recommendation: Please expand the discussion on the clinical application and generalizability of the findings. In particular, consider addressing how the results might translate to cytokine measurements obtained via methods other than Luminex

Tables

No comments

Author Response

Thank you very much for taking the time to review this manuscript. Please refer to the attached file for a detailed responses.

Reviewer 2 Report

Comments and Suggestions for Authors

Dear Authors,

I appreciate the opportunity to review your manuscript addressing the critical issue of identifying biomarkers in papillary thyroid carcinoma (PTC). Your exploration of cytokine profiling using xMAP technology to analyze 46 cytokines provides valuable insights and opens avenues for applying such methodologies in this field.

Introduction:

Your introduction offers a comprehensive overview of cytokine profiling in PTC and the application of machine learning algorithms in medical diagnostics. However, it could benefit from a more detailed discussion of prior cytokine studies in thyroid cancer to better contextualize the study's objectives. I recommend enhancing the introduction by reviewing existing literature on cytokine studies in thyroid cancer and highlighting the gaps that this study aims to address.

Methods:

The methods section is detailed, covering sample processing, cytokine analysis, and AI model implementation. However, it lacks specifics on hyperparameter tuning, feature selection, and the handling of missing data. Major concerns in this section include:

The manuscript does not specify critical machine learning parameters and steps.

It is unclear how hyperparameters for the models were selected beyond grid search.

The dataset was split into four outer folds for validation—why four and not a standard 5 or 10-fold cross-validation?

While nested cross-validation is used, there is no mention of feature selection or dimensionality reduction.

Please provide:

Details on how hyperparameters were optimized (e.g., learning rate, depth, regularization in XGBoost).

An explanation for choosing a four-fold split instead of other standard splits.

Information on whether feature selection or dimensionality reduction was performed.

Age-Related Differences:

To strengthen your findings, please consider analyzing age differences between patient and control groups, as age can influence cytokine levels and the tumor microenvironment in PTC. Studies have shown that older PTC patients often exhibit higher tumor mutational burden and alterations in immune cell composition, such as decreased cytotoxic CD8+ T cells and increased resting dendritic cells, contributing to a tumor-permissive environment. Additionally, age is a well-established prognostic factor in PTC, with patients over 65 years experiencing more aggressive disease and poorer survival rates. Addressing potential age-related differences would enhance the robustness of your conclusions.

Discussion:

Expand the discussion to elaborate on the clinical relevance of the identified cytokines and the potential integration of this approach into current diagnostic workflows.

Author Response

(The authors gave the same response as above.)

Reviewer 3 Report

Comments and Suggestions for Authors

The Mann–Whitney U test was used in this study, but no further information was given. What was the point of using it, and how did this random test help them?

What makes this model very different from other t-tests? Did you check for biases with other tests?

It looks good, that SHAP model. I think it would be helpful to show spread plots for all five classes, especially with the most important traits that the T-test found.

I'm referring to two medical studies that used AI: https://doi.org/10.3390/ijerph19116439 and https://www.jto.org/article/S1556-0864(23)01062-6/fulltext to lines 58–62 of the statement.

They used t-tests and common machine learning models to look at how well they could find people with papillary thyroid cancer (PTC), which is not something new.

State-of-the-art comparison is needed to show how significant this work is.

Author Response

(The authors gave the same response as above.)

Reviewer 4 Report

Comments and Suggestions for Authors

This study performed conventional statistics and machine learning of cytokines in serum of control and test patients of papillary thyroid cancer. The text is well written, it is easy to read and understand. The following comments may help the authors to improve the analysis and manuscript:

Comments:

(1) Lines 40-45. Please provide a more detailed description of the pathological mechanism. Please add a figure if necessary.

(2) Please describe the basics of papilary thyroid carcinoma (PTC), the clinical information, pathology, and treatment options.

(3) Is it possible to diagnosie PTC based only on the cytokine profilinig?

(4) Line 54. Please describe the "invasive method".

(5) Lines 58-67. Please describe difference between machine learning algorithms and artificial neural networks as the mathematical background is different, including the intepretation of results (and realiability and application to clinical medical field).

(6) Line 93. Regarding the freezer. Do you know the brand and model?

(7) Line 93. Are the normal and the PTC cases comparable? Was a technique like the propensity score matching performed?

(8) Line 97. Please add the last statement and website link for the 2024 Helsinki Declaration (75th WMA General Assembly, Helsinki, Finland, October 2024) (https://www.wma.net/policies-post/wma-declaration-of-helsinki/)

(9) Regarding 2.2. What type of normal samples were used? Voluntary people? Patients with other diseases?

(10) Section 2.3. Please add the catalog number and company of all reagents. You may add a table a appendix A.

(11) Line 123. PD-L1 is not a cytokine. Could you please explain why it was added?

(12) Line 127. Regarding "A minimum of 50 beads per analyte was acquired". Please explain the meaning of this sentence as not all readers may understand it properly.

(13) Line 131. Regarding "intensities below the background". Please describe with more detail.

(14) Does the freezing and warming affect the cytokine stability?

(15) Lines 148-153. Sorry to ask but, wouldn't be safer to exclude the cases with missing data? Is the strategy of replacing with minimum, maximum, and mean values correct, and would this introduce a bias? Of note, some algorithms cannot handle missing data, others will use missing data to classify the cases and may make a bias. Please discuss and reason the analysis strategy.

(16) Is the evaluation of cytokines performed in duplicate?

(17) Line 182. Please describe Grid Search method.

(18) What software or code was used for the machine learning methods describes in lines 179-200?

(19) Regarding Table 1. In theory there are 63 control and 63 PTC. However, the is missing data.
19.1 What percentage of cases had missing data?
19.2. What percentage of variables had missing data?
19.3. What is the effect of missing data, is it detrimental?
19.4. If you exclude cases/variable with missing data, how does the table change? Could you please show Table 1 without missing data?

(20) As I understand from Table 1, both control and test groups are comparable. Aren't they?

(21) Table 1. Please add the exact p value of the difference between the 2 groups regarding the variable glucose. Additionally, why PTC patients have higher glycemia?

(22) In Table 2. If you perform non-parametric test for all variables, does the resuls change?

(23) Regarding Table 2. If you perform binary logistic regression, backwald conditional, what are the most relevant variables that remain?

(24) Regarding p values of 0.000, please add exact value and/or <0.001.

(25) Line 259. Sorry to ask, what do you mean by "20 models per algorithm"? Did you repeated the test 20 times, therefore you can calculate intervals for the performance parameters? 

(26) Do the classification algorithms create a differen output each time?

(27) Could you please descrive the basics of SHAP analysis with more detail?

(28) As I understand, the most relevant markers were EGF and IL10 in SHAP analysis. Using conventional statistics the result was similar, wasn't it?

(29) Should Figure 5 be included in results section? Please correct to "parameters"

(30) Figure 6. Was variable "IP-10" analyzed previously? It does not appear in Table 2. Did you mean IL-10? Please confirm.

(31) Why EGF and IL-10 are associated with the pathogensis of PTC. What is the pathological mechanism?

Author Response

(The authors gave the same response as above.)

Round 2

Reviewer 4 Report

Comments and Suggestions for Authors

Thank you for the answers.

Author Response

Dear authors,

Thank you for submitting your work for consideration. The study meet scientific criteria. However, I would like to ask you to use metrics (i.e. sensitivity/specificity, instead of F1/precision/recall) that are more commonly used in the medical field. That would make your study more citations.

Please let us know if you have any question.

Best regards,
HsinYao Wang

Thank you very much for taking the time to review this manuscript and providing considerate and helpful comments.

As the academic editor commented, we revised the manuscript, as following:

(Page 1, Line 30-32 in revised manuscript)

Among the five classification algorithms evaluated, XGBoost demonstrated superior performance in terms of accuracy, precision, sensitivity (recall), specificity, F1-score, and ROC-AUC score.

(Page 5, Line 212-214 in revised manuscript)

Classification performance is assessed by accuracy, precision, sensitivity (recall), specificity, f1-score, and area under the curve-receiver operator characteristic (AUC-ROC) score.

(Page 9, Line 286-287 in revised manuscript)

Table 3 summarizes the performance of classification algorithms based on accuracy, precision, sensitivity (recall), F1-score, specificity and ROC-AUC score.

(Page 9-10, Table 3 in revised manuscript)

 Table 3. Model performance for classification algorithms.

Model

Accuracy

Precision

Sensitivity (Recall)

F1-score

Specificity

ROC-AUC score

XGBoost

0.913

(0.897, 0.928)

0.922

(0.896, 0.948)

0.913

(0.888, 0.938)

0.912

(0.897, 0.928)

0.920

(0.889, 0.950)

0.964

(0.950, 0.977)

k-NN

0.653

(0.625, 0.681)

0.658

(0.629, 0.686)

0.661

(0.604, 0.718)

0.652

(0.620, 0.684)

0.646

(0.592, 0.700)

0.703

(0.668, 0.738)

Log. Reg.

0.892

(0.870, 0.920)

0.900

(0.850, 0.946)

0.886

(0.829, 0.952)

0.890

(0.880, 0.920)

0.897

(0.862, 0.932)

0.950

(0.931, 0.971)

NB

0.841

(0.871, 0.913)

0.840

(0.870, 0.932)

0.847

(0.848, 0.925)

0.840

(0.868, 0.912)

0.835

(0.799, 0.872)

0.919

(0.933, 0.967)

SVM

0.866

(0.835, 0.896)

0.866

(0.841, 0.892)

0.882

(0.811, 0.918)

0.862

(0.828, 0.897)

0.867

(0.837, 0.896)

0.916

(0.893, 0.938)

Mean, 95% confidence interval for six measurements were presented for each model.

Abbreviations: k-NN, k-nearest neighbor; Log. Reg., Logistic Regression; NB, Naïve Bayes; SVM, Support Vector Machine; XGBoost, Extreme gradient boosting

(Page 10, Line 296-298 in revised manuscript)

Based on the median, as with the mean, XGBoost outperformed all other algorithms across all six metrics, followed by logistic regression.

(Page 10, Line 303-305 in revised manuscript)

Five performance measurements were calculated; Precision, Recall (Sensitivity), F1-score, Accu-racy, ROC-AUC score.

(Page 11, Line 322-325 in revised manuscript)

Notably, performance evaluation scores exceeding 0.8 across five metrics in Figure 5 were achieved using only the two features, EGF and IL-10. And with the two features, specificity showed an average value of 0.798.

(Page 12, Line 345-350 in revised manuscript)

As a result, the evaluation metrics—accuracy, precision, F1-score, sensitivity and specificity—were meaningful indicators of performance, with XGBoost showing the highest performance. A previous study that compared the predictive performance of algorithms including XGBoost, SVM, k-NN, and LR for hematoma expansion prediction on a balanced dataset found that XGBoost outperformed the other models with accuracy, precision, recall (sensitivity), and F1-score of 0.82, consistent with our findings (33).

(Supplementary Table S5)

Supplementary Table S5. Explained Variance by the Number of Data Dimensions

Algorithms

ROC AUC

Accuracy

F1-score

Sensitivity (Recall)

Precision

Specificity

k-NN

0.703

0.653

0.652

0.661

0.658

0.659

k-NN (reduced)

0.849

0.721

0.755

0.689

0.744

0.662

NB

0.919

0.841

0.840

0.847

0.840

0.835

NB (reduced)

0.934

0.873

0.861

0.876

0.850

0.867

Abbreviations: k-NN; k-nearest neighborhood algorithm, NB; Naïve Bayes Classifier
